# Matrix metalloproteinase 1 modulates invasive behavior of tracheal branches during entry into *Drosophila* flight muscles

Julia Sauerwald[1,2], Wilko Backer[1,2], Till Matzat[1,2†], Frank Schnorrer[3], Stefan Luschnig[1,2]*

[1]Institute for Zoophysiology, University of Münster, Münster, Germany; [2]Cluster of Excellence EXC 1003, Cells in Motion (CiM), Münster, Germany; [3]Aix Marseille University, CNRS, IBDM, Marseille, France

**Abstract** Tubular networks like the vasculature extend branches throughout animal bodies, but how developing vessels interact with and invade tissues is not well understood. We investigated the underlying mechanisms using the developing tracheal tube network of *Drosophila* indirect flight muscles (IFMs) as a model. Live imaging revealed that tracheal sprouts invade IFMs directionally with growth-cone-like structures at branch tips. Ramification inside IFMs proceeds until tracheal branches fill the myotube. However, individual tracheal cells occupy largely separate territories, possibly mediated by cell-cell repulsion. Matrix metalloproteinase 1 (MMP1) is required in tracheal cells for normal invasion speed and for the dynamic organization of growth-cone-like branch tips. MMP1 remodels the CollagenIV-containing matrix around branch tips, which show differential matrix composition with low CollagenIV levels, while Laminin is present along tracheal branches. Thus, tracheal-derived MMP1 sustains branch invasion by modulating the dynamic behavior of sprouting branches as well as properties of the surrounding matrix.
DOI: https://doi.org/10.7554/eLife.48857.001

*For correspondence:
luschnig@uni-muenster.de

Present address: †Wine Research Centre, The University of British Columbia, Vancouver, Canada

Competing interests: The authors declare that no competing interests exist.

## Introduction

Indirect flight muscles (IFMs) of flying insects display the highest known metabolic rates in the animal kingdom (*Weis-Fogh, 1964*). In *Drosophila*, two sets of IFMs, the dorsal-longitudinal muscles (DLMs) and the perpendicularly oriented dorso-ventral muscles (DVMs) are anchored to the thoracic cuticle and move the wings indirectly by deforming the thoracic exoskeleton rather than by acting directly on the wings. Each adult IFM is approximately 1 mm long and 100 μm wide (*Spletter et al., 2018*) and contains about 1000 nuclei (*Rai and Nongthomba, 2013*). To supply these large muscles with sufficient oxygen, an extensive network of gas-filled tracheal tubes not only superficially enwraps the IFMs, but also invades the myotube interior. This remarkable physiological adaptation minimizes the distance for oxygen diffusion from tracheoles to muscle mitochondria (*Weis-Fogh, 1964*; *Wigglesworth and Lee, 1982*) and provides efficient gas exchange for aerobic respiration to sustain flight over long time periods (*Götz, 1987*).

Tracheal cell migration is controlled by Fibroblast growth factor (FGF) signaling (*Ghabrial et al., 2003*; *Hayashi and Kondo, 2018*). The FGF ligand Branchless (Bnl) acts as a chemoattractant (*Sutherland et al., 1996*) that promotes tracheal cell motility by activating the receptor tyrosine-kinase (RTK) Breathless (Btl) on tracheal cells (*Klambt et al., 1992*). IFMs receive their tracheal supply from tracheal cells that extend from the thoracic air sac primordium towards the notum region of the wing imaginal disc during larval development (*Sato and Kornberg, 2002*). Subsequently, during

metamorphosis, tracheal terminal branches (tracheoles) ramify on and invade the developing IFMs (*Peterson and Krasnow, 2015*). Tracheal invasion into IFMs depends on the attraction of tracheal branches by Bnl FGF secreted on the muscle surface, followed by a switch to release of FGF from the interior transverse (T)-tubule system (*Peterson and Krasnow, 2015*). The T-tubule system is a network of tubular longitudinal and transversal membranes that extend around each sarcomere and are required for excitation-contraction coupling (*Razzaq et al., 2001*). It is continuous with the plasma membrane and was proposed to provide entry points for invasion of tracheal branches into the IFMs (*Peterson and Krasnow, 2015*). However, how tracheal cells interact with and enter the myotube, and how this process is coordinated with muscle development, has not been clear.

Tracheal invasion into IFMs presumably requires dynamic remodeling of extracellular matrix (ECM) and plasma membranes, but the underlying mechanisms are not well understood. Matrix metalloproteinases (MMPs) are involved in tissue reorganization during branching morphogenesis in various systems, including the mammalian lung (*Atkinson et al., 2005*; *Wiseman et al., 2003*), mammary gland (*Wiseman et al., 2003*), and the *Drosophila* tracheal system (*Page-McCaw et al., 2003*). The *Drosophila* genome encodes two MMPs, MMP1 and MMP2, which perform common and distinct functions during tissue remodeling (*Llano et al., 2002*; *Page-McCaw et al., 2007*). MMP1 was shown to be required for tracheal remodeling during larval growth (*Glasheen et al., 2009* ) and MMP2 for normal outgrowth of the air sac primordium (*Wang et al., 2010*). MMPs can be either secreted or membrane-tethered (*LaFever et al., 2017*; *Page-McCaw et al., 2007* ), and are thought to function mainly as enzymes cleaving ECM components. However, MMP-mediated proteolysis can also modulate signaling by processing growth factors such as TNF $\alpha$ and TGF $\beta$ (*English et al., 2000*; *Yu and Stamenkovic, 2000*), by regulating growth factor availability and mobility (*Lee et al., 2005*; *Wang et al., 2010*), or by cleaving growth factor receptors (*Levi et al., 1996*). MMP2 was shown to restrict FGF signaling through a lateral inhibition mechanism that maintains highest levels of FGF signaling in tracheal tip cells (*Wang et al., 2010*). Moreover, MMPs can regulate mammary gland development independently of their proteolytic activity (*Kessenbrock et al., 2013*; *Mori et al., 2013*).

To understand the mechanisms underlying tracheal invasion into IFMs, we analyzed the dynamics of the process in vivo. This revealed that tracheal cells invade IFMs directionally and migrate inside the myotubes with dynamic growth-cone-like structures at branch tips until tracheal branches fill the myotube volume. MMP1 activity is required in tracheal cells for normal invasive behavior and for the dynamic organization of growth-cone-like branch tips. We found that MMP1 remodels the Collagen IV-containing ECM around invading branch tips, suggesting that tracheal-derived MMP1 sustains branch invasion by modulating the properties of the surrounding matrix.

## Results

### Tracheae invade flight muscles in a non-stereotyped, but coordinated manner

To understand the mode of IFM tracheation, we first analyzed tracheal branch pathways on the surface of and within IFMs. We focused our analysis on DLMs, which receive their tracheal supply from thoracic air sacs (*Figure 1A*). Stochastic multicolor labeling of tracheal cells (*Nern et al., 2015*) revealed that multicellular air sacs converge into unicellular tubes (*Figure 1B*) with ramified tracheal terminal cells at their ends (*Figure 1B'*). Unlike tracheal terminal cells in other tissues, IFM tracheal cells not only ramify on the myotube surface, but also inside the syncytial myotube (*Figure 1C,C' and D,D'*; *Video 1*; *Peterson and Krasnow, 2015*). The cell bodies, including the nuclei, of IFM tracheal terminal cells reside on the myotube surface (*Figure 1C,C'' and D,D''*), while IFM nuclei are distributed throughout the muscle between myofibril bundles as well as near the muscle surface (*Figure 1C,C''' and D,D'''*).

In each IFM, branch invasion starts between myofibril bundles and fine subcellular tracheal branches (tracheoles) also invade myofibril bundles (*Figure 1D,D'*; *Video 1*), with most tracheoles extending parallel to the myofibrils (*Figure 1E,E'*). The number and morphology of terminal cells supplying a specific DLM was variable between individuals, indicating that the tracheal branching pattern in IFMs is not stereotyped (*Figure 1—figure supplement 1B*). Interestingly, however, the number of tracheal branches was relatively uniform along DLM myotubes (*Figure 1—figure*

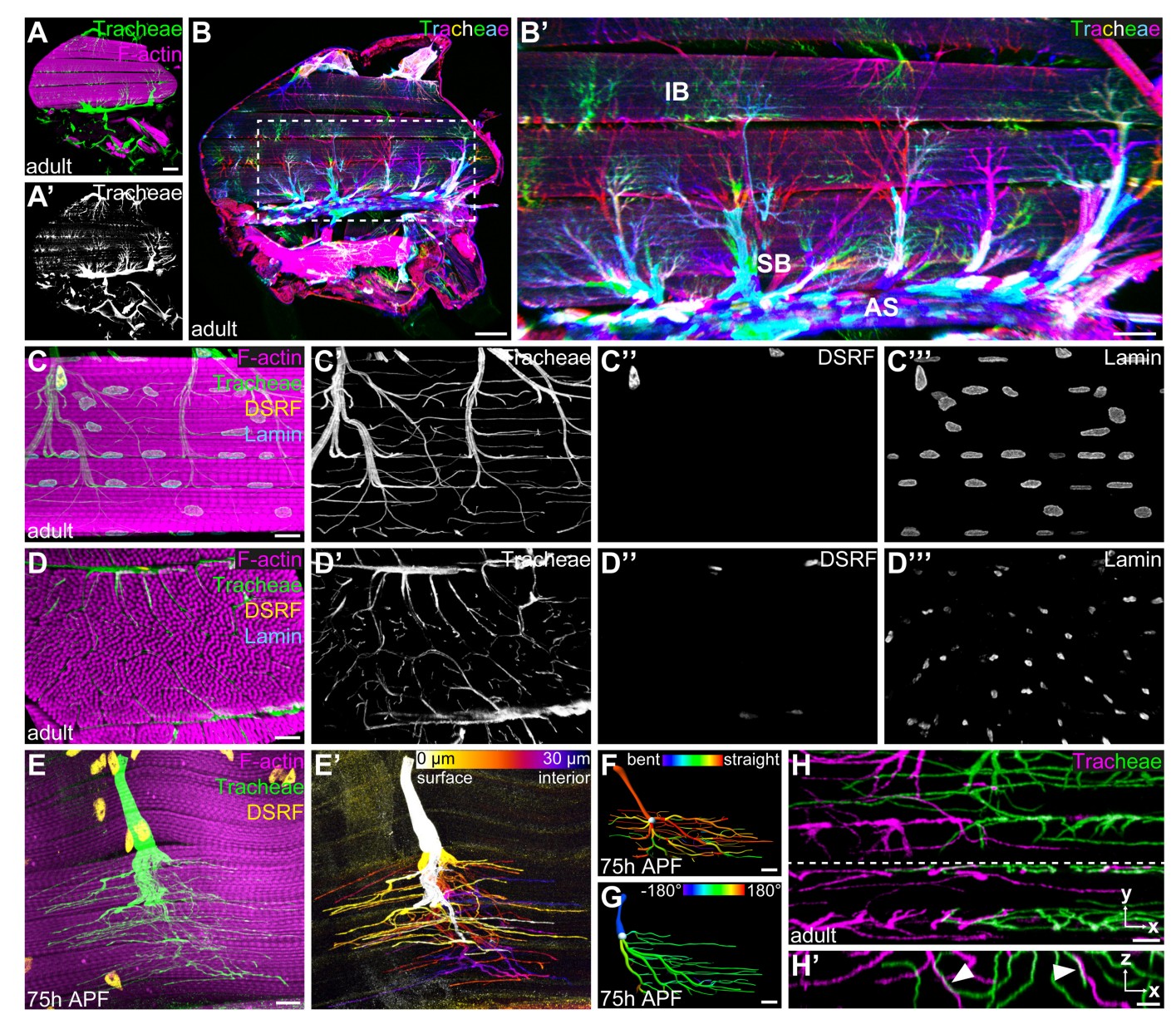

**Figure 1.** Tracheal terminal cell branches occupy separate territories in IFMs. (A–A') Sagittal section of an adult thorax with dorsal longitudinal muscles (DLMs) stained for F-actin (magenta). Tracheal branches, visualized by their autofluorescence (green) arise from the thoracic air sacs adjacent to IFMs. (B) Stochastic multicolor labeling of tracheal cells in a sagittal section of an adult thorax. Multicellular tubes (B') emanating from air sacs (AS) are superficial branches (SB) with tracheal terminal cells at their ends. Terminal cell branches spread on the muscle surface and invade as internal branches (IB) into the myotube. Note that individually labeled terminal cells occupy largely separate territories in IFMs (B'). (C,D) Tracheal branch supply of a single myotube in sagittal (C–C''') and cross- (D–D''') section stained for F-actin (magenta), LaminDm0 (cyan; all nuclei) and DSRF (yellow; tracheal terminal cell nuclei). Tracheal autofluorescence is shown in green. Cell bodies of tracheal terminal cells with DSRF-positive nuclei are located on the myotube surface. Terminal branches spread on the myotube surface, but also invade between and within myofibril bundles. (E) Single MARCM-labeled terminal tracheal cell in a developing IFM 75 hr APF. The tracheal cell, labeled with cytoplasmic GFP (green) and nuclear DSRF (yellow), extends fine tracheoles parallel to myofibrils (magenta). Color-coding of depth (E') indicates that a single tracheal cell ramifies deep into the myotube. The color map to the upper right indicates depth in the z-axis. (F,G) Segmented IFM tracheal terminal cell color-coded for branch straightness (F) and branch orientation angle (G). Note that the majority of branches extend straight projections (F) and that branches extending from a given terminal cell often display a bias towards one orientation along the myotube long axis (G). (H–H') Close-up of branches from two differentially labeled tracheal terminal cells (green and magenta). Note that while individual terminal cells occupy largely separate territories within the myotube, some branches of adjacent cells appear to be in close proximity or direct contact (arrowheads in H'). H' shows an orthogonal section in the x-z plane indicated by a dashed line in H. Scale bars: 100 µm (A, A',B), 50 µm (B'), 10 µm (C–G), 5 µm (H,H').

*Figure 1 continued on next page*

*Figure 1 continued*

DOI: https://doi.org/10.7554/eLife.48857.002

The following figure supplement is available for figure 1:

**Figure supplement 1.** Tracheal terminal cells with non-stereotyped cellular morphologies fill the myotube volume.

DOI: https://doi.org/10.7554/eLife.48857.003

*supplement 1A*, n = 6). These findings suggest that branches originating from tracheal terminal cells uniformly fill the available myotube volume in a manner that is non-stereotyped, but tightly coordinated with myotube morphology.

## Tracheal cells occupy separate territories within the myotube

To investigate how tracheal cells arrange within myotubes to fill their volume, we generated animals carrying individually marked tracheal cells. Morphometric analysis of 31 individual IFM terminal tracheal cells revealed that these cells display highly variable cellular architectures, as measured by cellular volume, sum of branch lengths, and the number of branch and terminal points (*Figure 1—figure supplement 1B*, *Supplementary file 1*). However, certain features were more uniform among cells. At least 95% of the branches of a given cell were aligned with the myotube axis (*Figure 1F*; n = 31) and the direction of branches was often biased towards one end of the myotube (*Figure 1G*, *Figure 1—figure supplement 1B*). Stochastic multicolor labeling revealed that individual tracheal terminal cells occupy largely non-overlapping territories (*Figure 1B'*). Interestingly, at the borders of such territories, branches from different cells were occasionally in close proximity or in direct contact (*Figure 1H,H'*). These findings suggest that invading tracheal cells fill the available space within the myotube, but minimize overlaps, possibly mediated by contact-dependent repulsion between tracheal cells.

## Innervation precedes tracheal invasion into IFM myotubes

To investigate the dynamics of IFM tracheation, we analyzed a time course between 32 hr APF and adulthood. While DVMs form de novo by fusion of adult muscle progenitors (AMPs), DLMs form by fusion of AMPs to larval 'template' muscles 8 hr APF (*Weitkunat et al., 2014*; *Dutta et al., 2004*; *Fernandes et al., 1991*). Tracheal invasion into DLMs begins around 48 hr APF when tracheoles start to enwrap and invade the myotubes (*Figure 2A,A'*; *Peterson and Krasnow, 2015*). Around 60 hr APF DLM tracheation is not complete yet, indicating that tracheal ramification inside DLM myotubes continues during late pupal development (*Figure 2B,B'*). Interestingly, prior to entry of tracheal branches into myotubes, motor neurons have already innervated IFMs (*Figure 2A,A''*). Furthermore, the distribution of tracheal branches was largely distinct from that of motor neurons (*Figure 2B, B''*) and the main tracheal branches did not overlap with motor neuron axons, suggesting that tracheae and neurons use separate entry routes into the muscle. Thus, tracheal invasion into myotubes is a comparatively slow process that occurs after IFM innervation.

## IFM mitochondria enwrap tracheal branches, but not vice versa

Classical studies using dye infiltration experiments (*Wigglesworth and Lee, 1982*) described that IFM tracheole endings encircle IFM mitochondria, suggesting that mitochondria may be involved in guiding tracheal invasion inside the muscle. To investigate how mitochondria might influence tracheal branch pathways we analyzed the interrelationship of mitochondria and tracheae during IFM development using

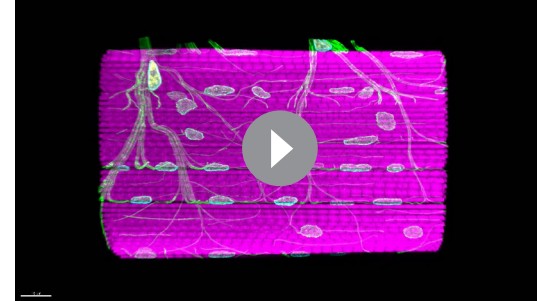

**Video 1.** Organization of tracheal branch invasion into a single IFM syncytium. 3D animation of z-stack of a single DLM stained for F-actin (magenta). Immunostaining against LaminDm0 (cyan) labels all nuclei, DSRF (yellow) labels tracheal terminal cell nuclei. Tracheal branches (green) were visualized by their autofluorescence.

DOI: https://doi.org/10.7554/eLife.48857.004

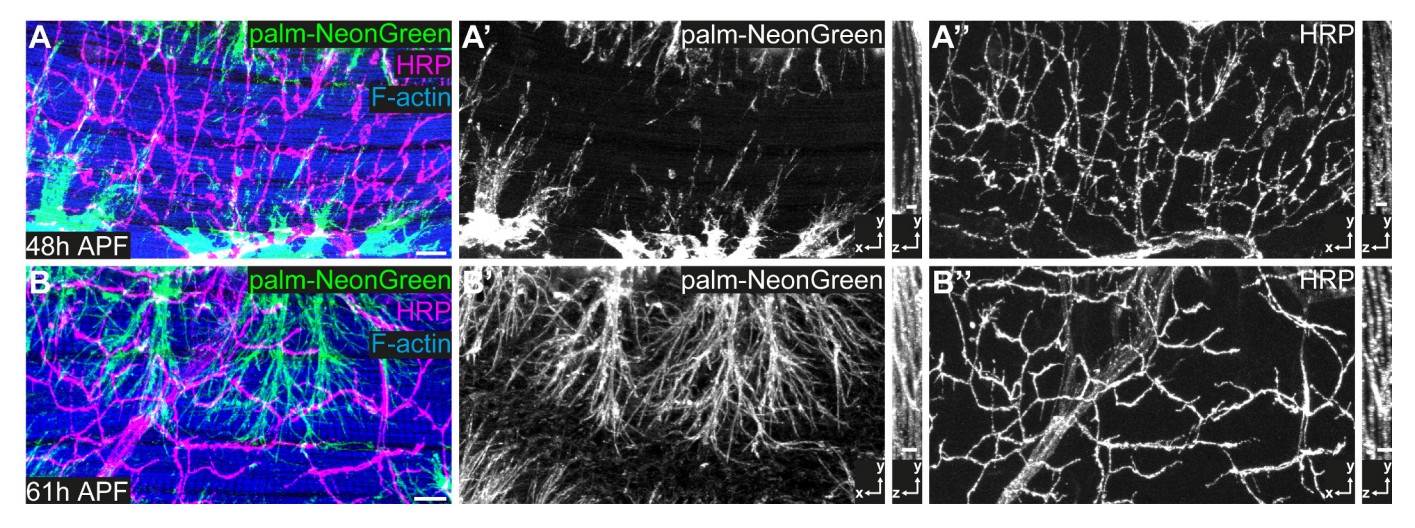

**Figure 2.** IFM innervation precedes tracheal invasion. (**A,B**) DLMs at 48 hr APF (**A**) and 61 hr APF (**B**). Tracheal cell membranes are labeled by palmitoylated mNeonGreen (green) driven by *btl*-Gal4. F-actin is labeled with Phalloidin (blue) and neurons with HRP (magenta). Note that at 48 hr APF (**A–A''**), when tracheal invasion has just started, the muscle is already innervated with motor neurons. At 61 hr APF tracheae have invaded the myotube (**B–B''**). Cross-sections to the right show that most tracheal branches reside on the surface of the myotube at 48 hr APF, whereas tracheal branches are inside the myotube at 61 hr APF. Scale bars: 10 μm (**A,B**).

DOI: https://doi.org/10.7554/eLife.48857.005

The following source data and figure supplements are available for figure 2:

**Figure supplement 1.** Mitochondria change their morphology during IFM development and partially enwrap tracheoles.

DOI: https://doi.org/10.7554/eLife.48857.006

**Figure supplement 1—source data 1.** Source data for *Figure 2—figure supplement 1I*.

DOI: https://doi.org/10.7554/eLife.48857.007

transmission electron microscopy (TEM). During sarcomere assembly in DLMs mitochondria change from a tubular morphology with few cristae to globular mitochondria with an elaborate cristae network (*Figure 2—figure supplement 1A–H*), accompanied by a dramatic increase in mitochondrial size between 48 hr APF and eclosion (*Figure 2—figure supplement 1I*). Adult IFMs are packed with globular mitochondria between myofibrils, yielding tracheal branches closely associated with mitochondria along their entire length. However, in contrast to earlier reports (*Wigglesworth and Lee, 1982*), we were unable to detect any cases in which tracheal branches encircled mitochondria. Strikingly, however, we found that some mitochondria partially enwrapped tracheal branches (*Figure 2—figure supplement 1J–K*; *Video 2*). These mitochondria showed no differences in volume or sphericity compared to mitochondria that were located farther away from tracheal branches (*Figure 2—figure supplement 1L,M*). Taken together, tracheal branches interact closely with mitochondria due to their dense packing between myofibrils and the partial enwrapping of tracheoles by mitochondria.

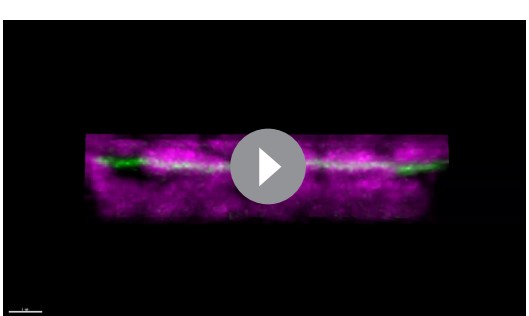

**Video 2.** Mitochondria can enwrap IFM tracheoles. 3D animation of an IFM tracheal branch surrounded by multiple mitochondria. Mitochondria were labeled by immunostaining against ATP5A (magenta). The tracheal branch (green) was visualized by its autofluorescence.

DOI: https://doi.org/10.7554/eLife.48857.008

## *salm*-dependent flight muscle fate is required for tracheal invasion

We used tissue-specific RNAi to systematically search for tracheal- and muscle-derived factors, respectively, required for IFM invasion (*Figure 3—figure supplement 1A*). As previously

reported (*Peterson and Krasnow, 2015*), the Bnl FGF chemoattractant is essential for IFM tracheation, as muscle-specific knock-down of Bnl completely abolished tracheal invasion into IFMs (*Figure 3—figure supplement 1B,C*). Interestingly, the trachealess muscles developed into adult IFMs with normal morphology of myofibrils, sarcomeres, and mitochondria, and with innervation by motor neurons (*Figure 3—figure supplement 1B,C,E–H*), suggesting that tracheal supply is dispensable for normal IFM development. However, adult flies lacking IFM tracheae were unable to fly. This finding prompted us to search for additional genes with roles in IFM tracheation using muscle-specific RNAi. We analyzed a set of 66 genes (*Supplementary file 2*), which are required in IFMs for muscle function (flight), but not for normal muscle morphology (*Schnorrer et al., 2010*), suggesting that these genes may be involved in IFM tracheation. We used *mef2*-Gal4 to knock down each of these genes in all muscles, and screened for changes in IFM tracheation. However, among the genes tested only the transcription factor Spalt major (Salm), which specifies fibrillar muscle fate (*Schönbauer et al., 2011*), was required for tracheal invasion into muscles (*Figure 3—figure supplement 1D*). Thus, as tracheal invasion only occurs in fibrillar muscles and not in other muscle types in *Drosophila* (*Peterson and Krasnow, 2015*), Salm-dependent processes appear to play a key role in preparing myotubes for tracheal invasion.

## MMP1 is required in tracheal cells for invasion into myotubes

We used an analogous RNAi approach to search for factors required in tracheal cells for branch invasion into myotubes (*Supplementary file 2*) and identified an important role of matrix metalloproteinase 1 (MMP1) in this process. Knock-down of *Mmp1* in tracheal cells using *btl*-Gal4 led to altered tracheal branching on IFMs (*Figure 3A–C′*). The angles between branches emanating from tracheal cell bodies on the myotube surface were reduced compared to control animals (*Figure 3E–G*). In addition, fewer tracheal branches were found inside myotubes (*Figure 3H,I,J*) and the fraction of myotube volume occupied by tracheoles was reduced (L). Furthermore, the tracheoles inside the myotubes showed fewer branch points compared to controls (*Figure 3M*). Consistent with these findings, tracheal *Mmp1* knock-down led to reduced flight ability, indicating compromised muscle function of adult flies (*Figure 3N*). We confirmed the specificity of the RNAi effect using two independent dsRNAs targeting different regions of the *Mmp1* gene, *Mmp1* RNAi 1 (JF01336, TRIP; *Figure 3B,B′,J,J′*) and *Mmp1* RNAi 2 (*Uhlirova and Bohmann, 2006*), which led to comparable tracheal phenotypes (*Figure 3L,M*). Furthermore, introducing the *Mmp1* homologue of *Drosophila pseudoobscura* under the control of its endogenous promoter (*Ejsmont et al., 2009*) into animals expressing *Mmp1* dsRNA in tracheal cells restored normal tracheal IFM invasion (*Figure 3C,C′,J,J′, L,M*). Together these results indicate that MMP1 is required in tracheal cells for invasion into myotubes and that the defects observed upon expression of *Mmp1* dsRNA are due to depletion of *Mmp1* (*Figure 3A–C*).

To assess the distribution of MMP1 in IFM tracheae we analyzed flies carrying a genomic fosmid clone in which MMP1 has been fused at its C-terminus to superfolder GFP and FLAG, TY1, and V5 epitope tags (fTRG145; *Sarov et al., 2016*). MMP1$^{fTRG145}$ signals were detectable by anti-GFP immunostaining on the IFM-associated tracheal air sac epithelium and along tracheal branches inside IFM myotubes (*Figure 4—figure supplement 1*), consistent with our finding that MMP1 is required for normal invasive behavior of tracheal branches in IFM myotubes.

Since Mmp1 has membrane-tethered and secreted isoforms (*LaFever et al., 2017*), Mmp1 could exhibit both cell-autonomous and cell-non-autonomous functions during branch invasion. To investigate Mmp1´s mode of action, we generated mosaic animals carrying clones of cells homozygous for the amorphic alleles *Mmp1$^{Q112}$* and *Mmp1$^2$* (*Page-McCaw et al., 2003*). However, we were not able to detect *Mmp1$^{Q112}$* or *Mmp1$^2$* mutant cells among IFM tracheae of mosaic animals (75 hr APF). Although we cannot exclude the presence of additional cell-lethal mutations on the *Mmp1* mutant chromosomes, the absence of homozygous *Mmp1* mutant clones from IFMs is consistent with an essential cell-autonomous requirement of MMP1 in IFM tracheation. Taken together, these findings indicate that MMP1 is required in tracheal cells for normal invasion into IFMs.

## Tracheal invasion depends on MMP1 proteolytic activity

To test whether MMP1 catalytic activity, rather than a non-catalytic function (e.g. of the MMP1 hemopexin domains), was required for branch invasion, we expressed the *Drosophila* tissue inhibitor

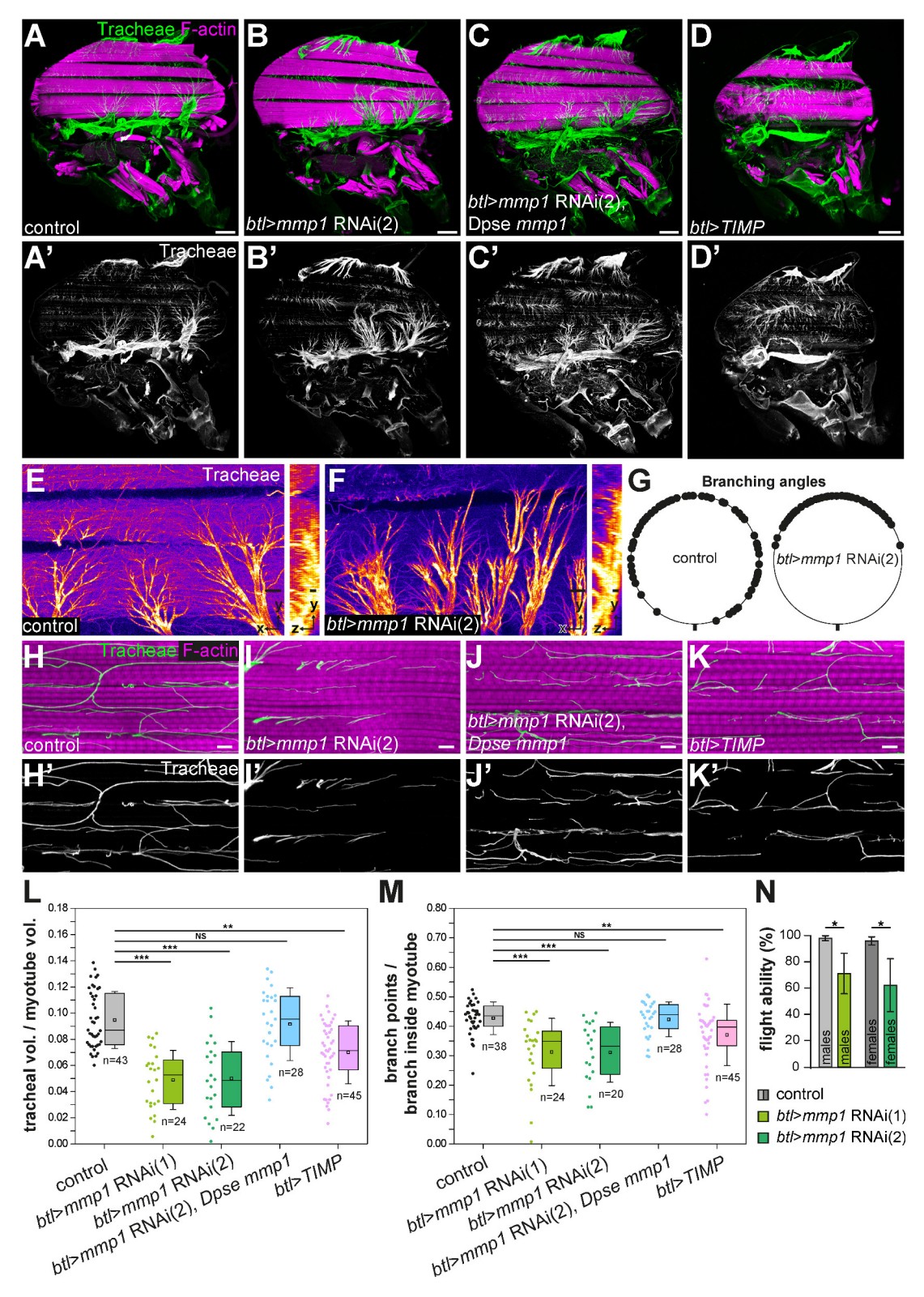

**Figure 3.** Tracheal branch invasion requires MMP1 function in tracheal cells. (A–D') Sagittal sections of adult thoraxes stained for F-actin (magenta). Autofluorescence of tracheae is shown in green. (E,F) show close-ups of tracheal branches on the IFM surface. Orthogonal sections (yz) are shown to the right. Note altered spreading of tracheal branches on the myotube surface upon tracheal-specific *Mmp1* knock-down. The effect of *Mmp1* knock-down was rescued by the *Drosophila pseudoobscura Mmp1* homologue *GA18484* (Dpse *Mmp1*; C). Tracheal expression of TIMP (D) phenocopies the

*Figure 3 continued on next page*

*Figure 3 continued*
effect of RNAi-mediated *Mmp1* depletion. (G) The angles between branches emerging from tracheal cell bodies on the myotube surface are reduced in tracheal *Mmp1* knock-down animals compared to controls. (H–K) Sagittal sections of single myotubes. Note reduced number of tracheoles inside myotubes upon tracheal *Mmp1* knock-down (I) compared to control (H). The effect of *Mmp1* knock-down was rescued by the *Drosophila pseudoobscura Mmp1* homologue GA18484 (Dpse *Mmp1*; J). Tracheal expression of TIMP (K) led to reduced tracheal invasion. (L,M) In a defined muscle volume the fraction occupied by tracheal branches (L) and the number of branch points per tracheal branch inside the myotube (M) were determined. At least 20 myotubes were scored. (N) Flight ability was measured as the percentage of flies that landed at the bottom immediately after throwing them into a Plexiglas cylinder (n = 4 experiments). Note that tracheal *Mmp1* knock-down leads to reduced flight ability due to impaired muscle function. *Mmp1* was depleted using *Mmp1* RNAi (1) or *Mmp1* RNAi (2) (*Uhlirova and Bohmann, 2006*). *p<0.05; **p<0.01; ***p<0.001; NS not significant. Scale bars: 100 µm (A–D'), 20 µm (E,F) and 5 µm (H–K').
DOI: https://doi.org/10.7554/eLife.48857.009
The following source data and figure supplements are available for figure 3:

**Source data 1.** Source data for *Figure 3L,M,N*.
DOI: https://doi.org/10.7554/eLife.48857.012
**Figure supplement 1.** Identification of genes required for branch invasion.
DOI: https://doi.org/10.7554/eLife.48857.010
**Figure supplement 1—source data 1.** Source Data for *Figure 3—figure supplement 1G,H*.
DOI: https://doi.org/10.7554/eLife.48857.011

of metalloproteinases (TIMP; *Pohar et al., 1999*) in tracheal cells under the control of *btl*-Gal4. TIMPs inhibit MMP activity by occupying the active site of the protease (*Gomis-Rüth et al., 1997*), and *Drosophila* TIMP was shown to inhibit MMP1 and MMP2 (*Page-McCaw et al., 2003*; *Wei et al., 2003*). *btl*-Gal4-driven expression of TIMP resulted in air sac defects (*Figure 3D,D'*), but adult flies were viable. The number of tracheal branches invading IFMs, as well as the number of tracheal branch points inside myotubes were reduced in these animals (*Figure 3K,K',L,M*), resembling the effect of tracheal *Mmp1* knock-down, although the defects caused by tracheal TIMP expression were milder compared to *Mmp1* knock-down. Together, these findings indicate that MMP catalytic activity is required in tracheal cells for normal IFM invasion.

## MMP1 modulates the speed of tracheal invasion into IFMs

To investigate the dynamics of IFM tracheal invasion and the function of MMP1 in this process, we developed a long-term live imaging protocol for visualizing IFM tracheation in living pupae (*Figure 4A*). Tracheal cell membranes were labeled with *btl*-Gal4-driven palmitoylated mKate2 (palm-mKate2) and muscles were labeled with Myofilin-GFP. We imaged the onset of IFM tracheal invasion at 48 hr APF, when tracheal branches extending from the air sac primordia begin to invade IFM myotubes (*Figure 4A,A'*). Tracheal branches extending from the medioscutal air sac invaded the myotube in a directional fashion from posterior to anterior along the myotube long axis (*Figure 4B,B'*; *Video 3* upper panel). Such directional invasion of tracheae was also apparent for tracheal branches that extend from the lateroscutal air sac and invade dorso-ventral IFMs (*Video 4* upper panel). Tracheal invasion upon tracheal-specific knock-down of *Mmp1* was still directional (*Video 4* lower panel), but the extent of tracheal ramification inside the myotubes at 62 hr APF was reduced compared to wild-type controls (*Figure 4*, compare B,B' to C,C'; *Video 3*). In control pupae the number of tracheal branches in a defined myotube volume close to the medioscutal air sac initially increased in a linear fashion and ceased at approximately 56 hr APF (*Figure 4D*; n = 5). In contrast, *Mmp1*-depleted tracheal cells entered at constant, but lower speed into the myotube (*Figure 4D*; n = 8). Thus, tracheal MMP1 function is required for the normal dynamics and speed of IFM tracheation (*Figure 4D'*).

## Invading tracheae display growth cone-like structures at branch tips

The altered dynamics of branch invasion upon tracheal *Mmp1* knock-down suggests that MMP1 proteolysis could be required for clearance of the entry path for tracheae into the myotube or for modulating signaling molecules that promote branch invasion. To elucidate the role of MMP1 we investigated the behavior of individual tracheal branch tips in wild-type and in tracheal MMP1 knock-down animals. Invading branches displayed growth cone-like structures at their tips, with dynamic protrusions resembling lamellipodia and filopodia (*Figure 4E*; *Video 5*). Growth-cone like structures

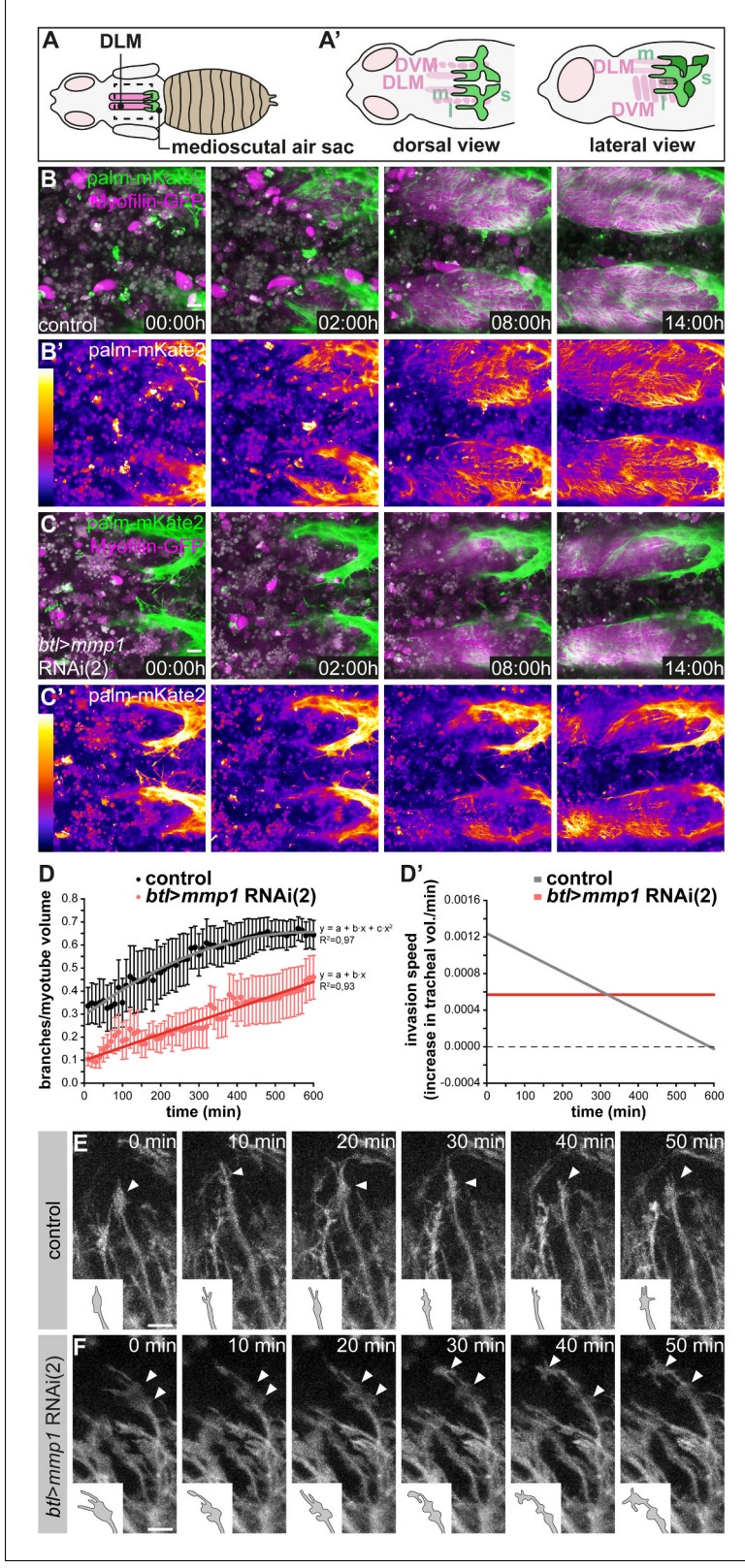

**Figure 4.** Normal dynamics of tracheal IFM invasion depends on tracheal Mmp1 function. (**A–A'**) Schematics of pupal flight muscles and air sacs 48 hr APF. The pupal case around the head and thorax was removed for live imaging of tracheal invasion into IFMs. Tracheal invasion into dorsal longitudinal muscles (DLMs) from the medioscutal (**m**) air sacs was imaged from a dorsal view (**A'**). Tracheal invasion into dorso-ventral muscles (DVMs)

*Figure 4 continued on next page*

*Figure 4 continued*

from the lateroscutal (I) air sacs was imaged from a lateral view (A'). (B,C) Stills of tracheal invasion into DLMs by branches arising from the medioscutal air sac in a control pupa (B,B') and a tracheal *Mmp1* knock-down pupa (C, C'). Palmitoylated mKate2 (palm-mKate2, green in B,C) labels tracheal cells, Myofilin-GFP (magenta in B,C) labels myotubes. B' and C' show palm-mKate2 intensities displayed as a heat map. The first time point (00:00 hr) corresponds to 48 hr APF. (D) Quantification of tracheal branches over time in a defined myotube volume close to the medioscutal air sac. The speed of invasion (increase in tracheal branch fraction per minute; D') was calculated for control and *Mmp1* knock-down pupae (n = 3). (E,F) Stills of tracheal branch tips invading DVMs in control (E) and in tracheal *Mmp1* knock-down pupa (F). Note that growth-cone like structures (arrowheads) are confined to branch tips in the wild type, but are found also along branch stalks upon *Mmp1* knock-down (n = 3). Scale bars: 20 µm (B,C) and 10 µm (E,F).

DOI: https://doi.org/10.7554/eLife.48857.013

The following source data and figure supplement are available for figure 4:

**Source data 1.** Source data for *Figure 4D*.
DOI: https://doi.org/10.7554/eLife.48857.015
**Figure supplement 1.** MMP1-GFP is distributed along tracheal branches in IFM myotubes.
DOI: https://doi.org/10.7554/eLife.48857.014

---

at branch tips were also observed upon tracheal *Mmp1* knock-down (*Figure 4F*; *Video 5*), and the presence of filopodia on these structures suggests that FGF signaling is active to promote migration also in cells with reduced MMP1 levels (*Ribeiro et al., 2002*). However, the growth-cone like structures in the MMP1-depleted cells did not remain confined to the branch tips as invasion proceeded (*Figure 4*, compare E to F). Instead, multiple enlarged protrusions resembling lamellipodia persisted, often behind the branch tip, suggesting that trachea-derived MMP1 is required for the dynamic organization of growth cone-like structures during tracheal migration inside the myotube.

## Extracellular matrix components are distributed non-uniformly along invading tracheal branches

The reduced speed of tracheal branch invasion upon *Mmp1* knock-down could arise from a defect in the ability of tracheal cells to clear their path through the surrounding BM, the removal or remodeling of which may require MMP1 function. To investigate potential roles of MMP1 in BM remodeling during branch invasion, we first analyzed the distribution of the BM components Laminin and Perlecan around tracheal branches at the onset of tracheal invasion 48 hr APF. Compared to adult IFM tracheae, which were covered with Laminin- and Perlecan-containing BM (*Figure 5—figure supplement 1C–C''*), invading tracheal branches at 48 hr APF showed little detectable BM (*Figure 5—figure supplement 1A–B'*). The muscle surface, however, was covered with Laminin and Perlecan (*Figure 5—figure supplement 1A–B'*). Of note, these ECM components also lined membrane invaginations on the myotube surface (*Figure 5—figure supplement 1D*), presumably representing openings of the T-tubule network (*Peterson and Krasnow, 2015*). Entry of tracheal branches into these invaginations could require MMP activity. However, tracheal *Mmp1* knock-down did not notably affect the levels and distribution of Perlecan around branch tips, and *Mmp1*-depleted tracheal cells invaded the myotube via the

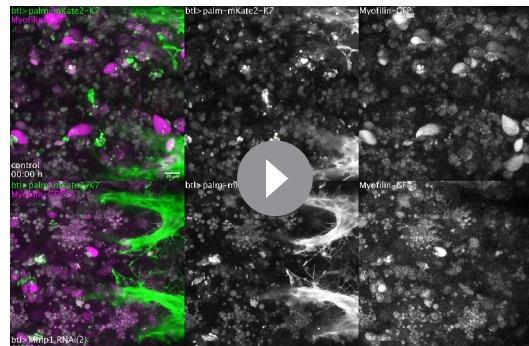

**Video 3.** Dorsal view of tracheal invasion into DLMs in control and tracheal-specific *Mmp1* knock-down animals. Time-lapse movies of tracheal invasion into DLMs in pupae 48 h APF. Tracheal cells are labeled by palmitoylated mKate2, IFMs are labeled by Myofilin-GFP. Dorsal views of a wild-type control pupa (top) and a tracheal *Mmp1* knock-down pupa (bottom) are shown. The movies were acquired with a 40x objective and a frame rate of 10 min in resonant scanning mode (Leica SP8) over 14 h. Z-stacks of ~100 µm (0.35 µm step size) were acquired.
DOI: https://doi.org/10.7554/eLife.48857.016

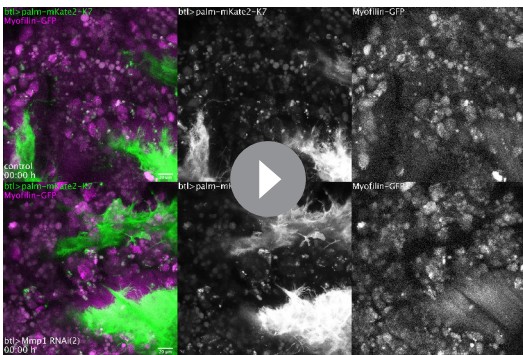

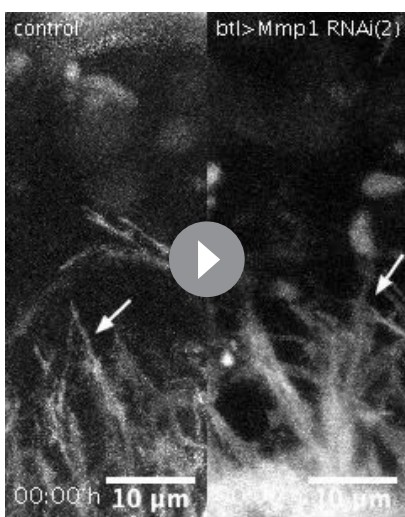

**Video 4.** Lateral view of tracheal invasion into DVMs in control and tracheal-specific *Mmp1* knock-down animals. Time-lapse movies of tracheal invasion into DVMs in pupae 48 h APF. Tracheal cells are labeled by palmitoylated mKate2, IFMs are labeled by Myofilin-GFP. Lateral views of a wild-type control pupa (top) and a tracheal *Mmp1* knock-down pupa (bottom) are shown. The movies were acquired with a 40x objective and a frame rate of 10 min in resonant scanning mode (Leica SP8) over 14 h. Z-stacks of ~100 μm (0.35 μm step size) were acquired.
DOI: https://doi.org/10.7554/eLife.48857.017

**Video 5.** Close-up of invading tracheal branch tips in control and tracheal-specific *Mmp1* knock-down animals. Close-up movies of tracheal invasion into DVMs in pupae 53 h APF. Tracheal cells are labeled by palmitoylated mKate2. A wild-type control (left) and a tracheal *Mmp1* knock-down pupa (right) are shown. Arrowheads point at growth cone-like structures. The movies were acquired with a 40x objective and a frame rate of 10 min in resonant scanning mode (Leica SP8). Z-stacks of ~100 μm (0.35 μm step size) were acquired.
DOI: https://doi.org/10.7554/eLife.48857.018

membrane invaginations (*Figure 5—figure supplement 1E–F'*).

Next, we analyzed the composition of the BM around tracheal branches in adult IFMs (*Figure 5*; *Video 6*). In controls Laminin was detected along the entire length of tracheal branches, both on the muscle surface and inside myotubes (*Figure 5A,A'',B,B''*; *Video 6*). In contrast, Perlecan and Collagen IV covered the tracheal stalks between myofibril bundles, but were not detectable around the tracheal tip regions inside myofibril bundles (*Figure 5A,A',B,B'*; *Video 6*), indicating that the BM around invading branch tips has a distinct composition and may be thinner than the BM surrounding the branch stalks.

## MMP1 is involved in remodeling of collagen IV around tracheal branches inside IFM myotubes

We asked whether tracheal-derived MMP1 influences the distribution of BM membrane components associated with tracheal branches. Depletion of tracheal MMP1 did not appear to affect the levels and distribution of trachea-associated Perlecan (*Figure 5C–D'*). However, *Mmp1* knock-down had a distinct effect on the distribution of Collagen IV. Whereas tracheal branch tips inside myotubes were devoid of Collagen IV in wild-type controls, 13% (n = 59) of *Mmp1*-depleted tracheal branch tips showed Collagen IV signals around the tip region (*Figure 5E–F'*). These findings suggest that trachea-derived MMP1 promotes invasion of tracheal branches through clearance of Collagen IV during migration inside the myotube.

## Discussion

Perfusion of tissues by oxygen-transporting vessels is a key prerequisite for all body functions in animals. In flying insects the extreme energy demands of flight are met by a network of tracheal tubes that minimize the distance for oxygen diffusion to mitochondria by extending branches into the interior of the flight muscles. This requires a new developmental process that enables tracheal cells to invade and spread throughout the IFM myotubes, unlike all other insect muscle types, where tracheoles ramify on the muscle surface only. Hence, flight muscle tracheation provides a powerful model to study the interactions that promote invasion of sprouting branches into tissues.

To investigate the cellular and molecular mechanisms underlying tracheal invasion into IFMs, we analyzed the dynamics of the process in vivo. First, through live imaging of muscle tracheation, we found that tracheal cells invade the muscle directionally with growth cone-like structures at branch tips. Tracheoles ramify inside the muscle until they uniformly fill the myotube volume. Intriguingly, however, single-cell analyses revealed that individual IFM tracheal cells occupy largely separate territories within the myotube, reminiscent of neuronal dendritic tiling (*Grueber and Sagasti, 2010*), suggesting that IFM tracheation involves repulsion between tracheal cells. Second, using a tissue-specific RNAi-based approach to identify factors required for branch invasion, we found that MMP1 activity is required in tracheal cells for normal speed of invasion and for the dynamic organization of growth-cone-like structures at migrating branch tips. Third, we showed that ECM components are distributed non-uniformly along IFM tracheal branches, with Laminin covering the entire length of tracheal branches, whereas Perlecan and Collagen IV are excluded from the tracheal tip regions inside the myotube. Fourth, we showed that MMP1 is involved in remodeling the Collagen IV-containing matrix around invading branch tips. Together, these findings suggest that MMP1 sustains tracheal branch invasion by modulating the dynamic behavior of tracheal branch tips and by remodeling the surrounding ECM.

A unique aspect of IFM tracheation is the fact that tracheal terminal cells enter and ramify within another cell, in this case a syncytial muscle. Although this system represents a specialized adaptation towards the extensive oxygen demand of this tissue, the underlying cellular mechanisms may be relevant also for the development of other organs, such as the vasculature. Tracheal invasion involves dynamic adhesion to the substrate, guidance of tracheal cells, and remodeling of the myotube ECM and plasma membrane to accommodate the invading tubes. Angiogenesis, which is based on tip-cell-guided migration with invasive protrusions probing the environment, involves similar challenges, for instance in the case of blood vessels that grow into collagen-packed cornea tissue or into bones (*Sivaraj and Adams, 2016*).

The ability of tracheal cells to enter the IFM myotubes is likely to depend on permissive and instructive cues provided by the muscle, as well as on factors that act in the tracheal cells to mediate their invasive behavior. We showed that tracheal invasion into IFMs critically depends on the transcription factor Salm, which specifies the fibrillar muscle type (*Schönbauer et al., 2011*). The *salm*-dependent cell fate switch appears to induce a program that renders the myotube permissive for tracheal invasion, for example through modulating properties of the muscle plasma membrane or ECM. In addition, Salm may regulate factors that mediate the dynamic redistribution of the Bnl FGF chemoattractant from the muscle surface to the internal T-tubule network in IFMs. This switch in the mode of the subcellular pathway of FGF secretion was shown to guide tracheal cells into IFMs (*Peterson and Krasnow, 2015*).

Classical electron microscopy studies suggested that tracheoles enter the IFMs through plasma membrane invaginations that are continuous with T-tubules, and then spread through the T-tubule network (*Smith, 1961a*; *Smith, 1961b*; *Wigglesworth and Lee, 1982*). Other muscle types that lack these membrane invaginations are not invaded by tracheal branches (*Peterson and Krasnow, 2015*). Surprisingly, however, we did not find evidence that a normally organized T-tubule system is required for tracheal ingrowth and spreading in *Drosophila* IFMs, since *amphiphysin (amph)* mutants with a disorganized T-tubule system (*Razzaq et al., 2001*) showed a normal distribution of tracheoles in IFMs (*Figure 5—figure supplement 1G,H*). Although the exact topology of the T-tubule system in wild-type and in *amph* mutant IFMs remain to be characterized, these results suggest that invasion into and spreading of tracheal cells inside IFMs may not depend on a pre-existing regular membrane invagination system.

IFM tracheoles are closely associated with mitochondria, thus minimizing the distance for gas exchange via diffusion. While we confirmed this close association by electron and high-resolution confocal microscopy, we found, contrary to an earlier report (*Wigglesworth and Lee, 1982*), no evidence that IFM tracheole endings encircle mitochondria. These earlier observations were based on dye infiltration experiments, which may be prone to artifacts due to leakage of the injected dye used for tracheal staining. Conversely, we discovered mitochondria that were partially enwrapping IFM tracheoles. This is likely due to extensive fusion of mitochondria, resulting in giant sleeve-like mitochondrial geometries around tracheal tubes in IFMs. Intriguingly, this arrangement is reminiscent of the mitochondria that wrap around the axoneme in sperm tails (*Woolley, 1970*). Thus, mitochondrial

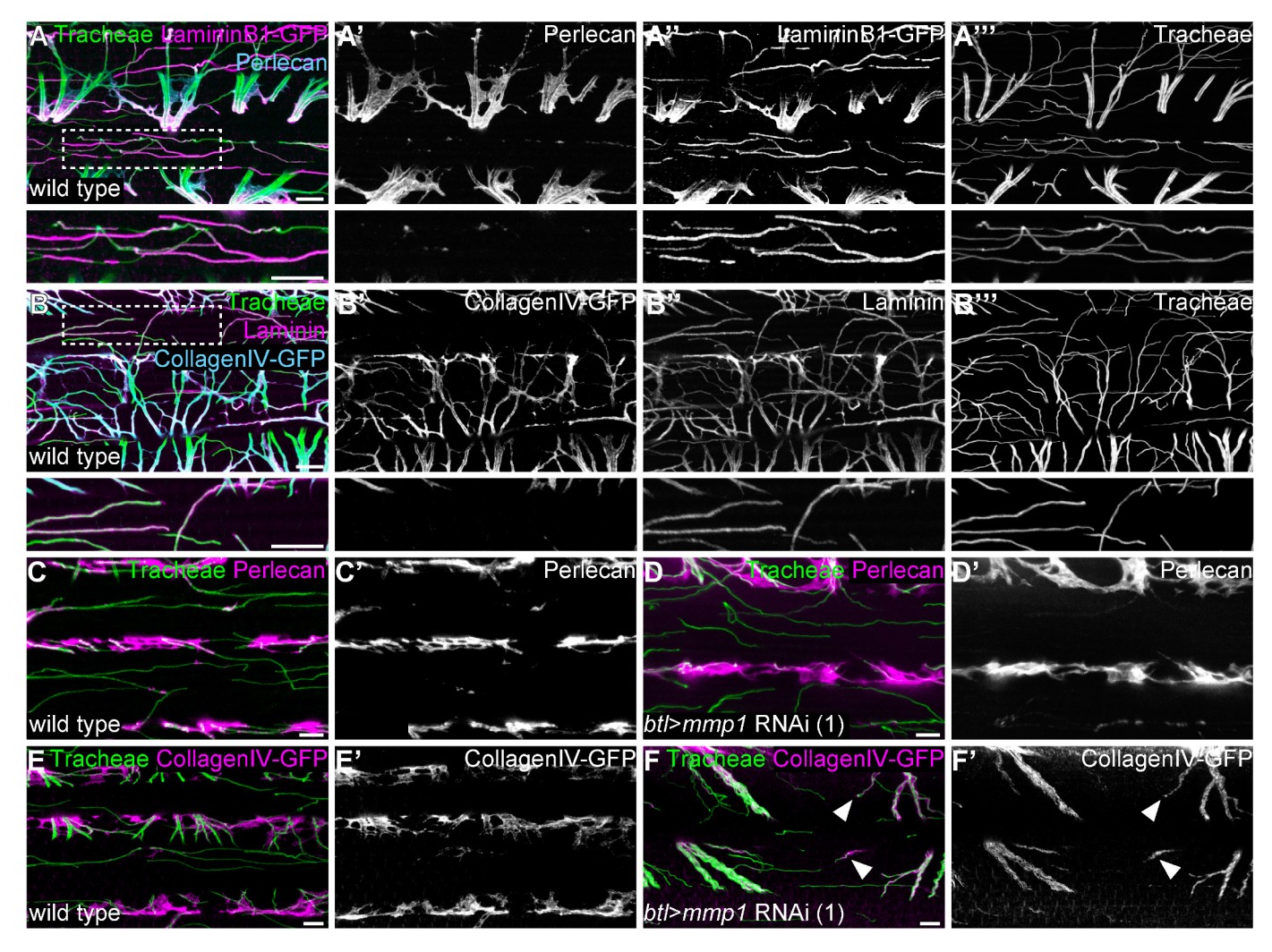

**Figure 5.** Tracheal stalk and tip regions show distinct basement membrane compositions. (A,B) Distribution of the ECM components Perlecan (A,A'), CollagenIV-GFP (B,B') and Laminin or LamininB1-GFP (A,A'',B,B'') around tracheal branches inside an adult IFM myotube. Note that Laminin extends along the entire length of tracheal branches, whereas Perlecan and Collagen IV are excluded from branch tips inside myofibril bundles. Bottom panels show close-up view of the regions marked by dashed boxes. (C–F) Distribution of Perlecan (C–D') and CollagenIV-GFP (E–F') around tracheal branches in wild-type control (C,C',E,E') and *Mmp1* tracheal knock-down (D,D',F,F') adult myotubes. Note that Perlecan distribution is not affected by *Mmp1* knock-down, whereas CollagenIV-GFP extends into the tip region of some branches in *Mmp1* knock-down, but not in control animals (F,F' arrowheads). *Mmp1* was depleted using *Mmp1* RNAi (1). Scale bars: 7 µm (A–B'''), 5 µm (C–F').

DOI: https://doi.org/10.7554/eLife.48857.019

The following figure supplement is available for figure 5:

**Figure supplement 1.** Basement membrane around tracheal branches increases during IFM development.

DOI: https://doi.org/10.7554/eLife.48857.020

wrapping may represent a common mechanism to sustain the extensive energy demands of specialized motile cell types such as flight muscle or sperm.

## MMP1 modulates invasive behavior of IFM tracheal cells

In addition to the muscle-derived factors discussed above, we show that the matrix metalloproteinase MMP1 is required in IFM tracheal cells for their normal invasive behavior. ECM remodeling is crucial for branching morphogenesis of various organs, and MMPs are the main enzymes that mediate ECM degradation (*Bonnans et al., 2014*). We showed that while the BM around invading

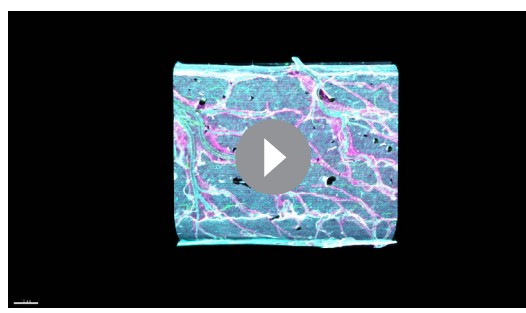

**Video 6.** Distribution of ECM around single adult DLM with tracheae. 3D animation of ECM around a single adult DLM with tracheae. ECM components Laminin (magenta) and Perlecan (cyan) were visualized by immunostaining. Tracheal branches (green) were visualized by their autofluorescence. Note that tracheae enter the myotube through ECM-lined invaginations of the muscle surface.

DOI: https://doi.org/10.7554/eLife.48857.021

branches is very thin at the beginning of IFM tracheation at 48 hr APF, tracheal branches in adult IFMs are surrounded by abundant, but molecularly heterogeneous, BM along their length. Laminin, but not Perlecan and Collagen IV, is present around branch tips. The presence of this molecularly distinct BM at the invasive branch tips suggests a reduced stiffness and increased distensibility of the BM, which has been observed also in other invading epithelia, including the mammalian salivary gland (*Bernfield et al., 1972*; *Grobstein and Cohen, 1965*; *Harunaga et al., 2014*), mammary gland (*Fata et al., 2004*) and lung (*Moore et al., 2005*). We found that depletion of MMP1 from tracheal cells led to a distinct effect on ECM remodeling. The tips of MMP1-depleted tracheal cells in mature IFMs displayed residual Collagen IV, suggesting that MMP1 is involved in modulating the mechanical properties of the ECM surrounding invading tracheal branch tips by removing CollagenIV-containing ECM. The moderate strength of this effect in MMP1 knock-down animals may be attributable to incomplete RNAi-mediated depletion or genetic redundancy of MMP1 with other matrix-degrading proteases. Furthermore, we cannot rule out that MMP1 might act on other ECM components, besides Collagen IV, that we did not test.

Since MMP1 has membrane-tethered and secreted isoforms (*LaFever et al., 2017*), MMP1 may execute both cell-autonomous and cell-non-autonomous functions. However, our results based on genetic mosaic analysis suggest that MMP1 acts in a cell-autonomous manner during IFM invasion. This function relies largely on MMP proteolytic activity, as we showed by expressing the MMP activity inhibitor TIMP. These findings raise the question as to which are the relevant substrates of MMP1 activity.

Historically, MMPs have been mainly associated with ECM remodeling (*Bonnans et al., 2014*). However, MMPs have broad substrate specificity and can cleave, besides several ECM components, also non-ECM proteins. Directional migration of tracheal cells in the embryo is mediated by the graded distribution of the Bnl FGF chemoattractant, which is expressed locally in small clusters of cells (*Sutherland et al., 1996*). In case of the developing IFMs, the redirection of FGF secretion from the cell surface to T-tubules explains the switch from superficial to invasive tracheal cell migration (*Peterson and Krasnow, 2015*). Yet, it is not clear whether and how an FGF chemotactic gradient could be established along a syncytial muscle to control directional persistence of tracheal cell migration along the IFMs. Since MMP1 is expressed in the air sac primordium (*Wang et al., 2010*), tracheal cell-associated MMP1 might influence the spatial distribution of the FGF chemoattractant by degrading Bnl FGF and thereby generating a local sink of chemoattractant around the invading branches. Localized degradation of the FGF chemoattractant by the migrating branch tips could sustain motility of sprouting branches towards areas with higher concentrations of FGF, even if FGF is initially distributed uniformly on the muscle surface. Although we have not been able to visualize the distribution of endogenous Bnl FGF on pupal IFMs due to the limited sensitivity of the available tools, analogous chemotactic gradients generated by migrating cells have been described in different developmental processes (*Donà et al., 2013*; *Tweedy et al., 2016*; *Venkiteswaran et al., 2013*). A regulatory interplay between MMPs and FGFs has been reported to operate also in other contexts of branching morphogenesis. MMP2-expressing tracheal tip cells are part of a lateral inhibition mechanism during larval *Drosophila* air sac development (*Wang et al., 2010*). Here, cells at the tip of the air sac primordium receive highest levels of FGF signaling and induce ERK-dependent expression of genes, including *Mmp2* (*Wang et al., 2010*). MMP2 mediates release of an inhibitory signal that acts non-autonomously to prevent FGF signaling in neighboring cells and consequently restricts tip cell fate to the MMP2-expressing cells. The nature of the inhibitory signal is still unknown. Interestingly, expression of MMP2 can be induced by FGF2 in mammalian endothelial cells

(*Kohn et al., 1995*). Mammalian MMP2 can also cleave FGFR1 to release a soluble receptor ectodomain fragment, which retains the ability to bind FGF and may influence FGF availability in the vascular BM (*Levi et al., 1996*). Together with our results, these findings suggest that MMP proteolytic activity may play conserved roles in different developmental contexts by controlling the invasive behavior of migrating cells through remodeling the properties of the surrounding matrix as well as by regulating growth factor signaling.

# Materials and methods

**Key resources table**

| Reagent type (species) or resource | Designation | Source or reference | Identifiers | Additional information |
|---|---|---|---|---|
| Genetic reagent (*Drosophila melanogaster*) | mef2-Gal4 | *Ranganayakulu et al., 1995* Dev Biol, 171(1), 169–181 | | |
| Genetic reagent (*Drosophila melanogaster*) | btl-Gal4 | *Shiga et al., 1996* Development, Growth and Differentation, 38(1), 99–106 | | |
| Genetic reagent (*Drosophila melanogaster*) | UAS-palmKate2-K7 | *Caviglia et al., 2016* Nat Cell Biol 18: 727–739 | | |
| Genetic reagent (*Drosophila melanogaster*) | UAS-palm-mNeonGreen | *Sauerwald et al., 2017* Development 144: 657–663 | | |
| Genetic reagent (*Drosophila melanogaster*) | UAS-FRT$_{FRT>STOP>FRT}$ myr::smGFP-HA_V5_FLAG | *Nern et al., 2015* PNAS 112: E2967–76 | | |
| Genetic reagent (*Drosophila melanogaster*) | $y^1$ $v^1$;; UAS-Mmp1_RNAi_(1) | Bloomington Stock Center | BDSC: 31489 | |
| Genetic reagent (*Drosophila melanogaster*) | UAS-Mmp1_RNAi_(2) | *Uhlirova and Bohmann, 2006* EMBO J 25(22): 5294–5304 | | |
| Genetic reagent (*Drosophila melanogaster*) | ;; FlyFos066598_DpseMmp1_GA18484 (attP-9A/VK13) | this study | | Mmp1 ortholog GA18484 from *Drosophila pseudoobscura* in genomic FlyFos vector inserted into VK13 attP landing site (76A2). See "Drosophila strains and genetics "in the Materials and methods section. |
| Genetic reagent (*Drosophila melanogaster*) | FRTG13 mmp1$^{Q273}$/CyO, Dfd-GMR-YFP | this study | | See "Genetic labeling of tracheal cell clones "in the Materials and methods section |

*Continued on next page*

*Continued*

| Reagent type (species) or resource | Designation | Source or reference | Identifiers | Additional information |
|---|---|---|---|---|
| Genetic reagent (*Drosophila melanogaster*) | *FRTG13 mmp1$^{Q112}$/CyO, Dfd-GMR-YFP* | this study | | See "Genetic labeling of tracheal cell clones "in the Materials and methods section |
| Genetic reagent (*Drosophila melanogaster*) | *FRTG13 mmp1$^2$/CyO, Dfd-GMR-YFP* | this study | | See "Genetic labeling of tracheal cell clones "in the Materials and methods section |
| Genetic reagent (*Drosophila melanogaster*) | *LanB1-GFP* | Vienna Drosophila Resource Center; *Sarov et al., 2016* | VDRC: fTRG681 | |
| Genetic reagent (*Drosophila melanogaster*) | *Mmp1-GFP* | Vienna Drosophila Resource Center; *Sarov et al., 2016* | VDRC: fTRG145 | |
| Genetic reagent (*Drosophila melanogaster*) | *Myofilin-GFP* | Vienna Drosophila Resource Center; *Sarov et al., 2016* | VDRC: fTRG501 | |
| Genetic reagent (*Drosophila melanogaster*) | *Vkg-GFP* | *Buszczak et al., 2007* *Genetics* 175: 1505–1531 | | |
| Genetic reagent (*Drosophila melanogaster*) | *UAS-TIMP* | Bloomington Stock Center | BDSC: 58708 | |
| Genetic reagent (*Drosophila melanogaster*) | *amph$^{26}$* | Bloomington Stock Center | BDSC: 6498 | |
| Antibody | Rabbit polyclonal anti-Bnl | *Jarecki et al., 1999* Cell, 99(2): 211–220 | | 1:250 |
| Antibody | Mouse monoclonal anti-Mmp1 | Developmental Studies Hybridoma Bank | Mix of mouse monoclonal antibodies 3B8D12, 3A6B4, 5H7B11 | 1:10/1:100/1:100 |
| Antibody | Mouse monoclonal anti-Dlg1 | Developmental Studies Hybridoma Bank | 4F3 | 1:200 |
| Antibody | Mouse monoclonal anti-DSRF | *Samakovlis et al., 1996* Development 122(5): 1395–1407 | mAb 2–161 | 1:100 |
| Antibody | Mouse monoclonal anti-ATP5A | Abcam | ab14748 | 1:100 |
| Antibody | Chicken polyclonal anti-GFP | Abcam | ab13970 | 1:500 |
| Antibody | Chicken polyclonal anti-HA | Abcam | ab9111 | 1:200 |
| Antibody | Mouse monoclonal anti-FLAG (M2) | Sigma | M2; Sigma-F1804 | 1:1000 |

*Continued on next page*

*Continued*

| Reagent type (species) or resource | Designation | Source or reference | Identifiers | Additional information |
|---|---|---|---|---|
| Antibody | Guinea pig polyclonal anti-LaminDm0 | gift from Georg Krohne, University of Würzburg | | 1:1000 |
| Antibody | Rabbit polyclonal anti-Laminin | *Schneider et al., 2006* Development 133(19): 3805–3815 | | 1:1500 |
| Chemical compound, drug | Phalloidin-TRITC | Sigma | | 1:1000 |
| Recombinant DNA reagent | fosmid clone FlyFos066598 | *Ejsmont et al., 2009* Nat Methods 6 (6): 435–437 | FlyFos066598 | fosmid clone FlyFos066598 containing the *Drosophila pseudoobscura* Mmp1 homologue GA18484 |

## *Drosophila* strains and genetics

The following *Drosophila* stocks are described in FlyBase and were obtained from the Bloomington stock center, unless noted otherwise: *btl*-Gal4 (*Shiga et al., 1996*), *mef2*-Gal4 (*Ranganayakulu et al., 1995*), UAS-*palm-mKate2* (*Caviglia et al., 2016*), UAS-*palm-mNeonGreen* (*Sauerwald et al., 2017*), UAS-*FRT>STOP>FRT-myr::smGFP-HA_V5_FLAG* (*Nern et al., 2015*), UAS-*Mmp1* RNAi (*Uhlirova and Bohmann, 2006*), $Mmp1^{Q273}$, $Mmp1^{Q112}$, $Mmp1^2$ (*Glasheen et al., 2009*), $amph^{26}$ (*Razzaq et al., 2001*), Mmp1-GFP (fTRG145), Myofilin-GFP (fTRG501), LamininB1-GFP (fTRG681; *Sarov et al., 2016*), Vkg-GFP (G205; *Buszczak et al., 2007*). Additional UAS-RNAi stocks were obtained from the TRiP or VDRC collections (*Supplementary file 2*). Crosses for RNAi knock-down were performed at 27°C. Heterozygous animals carrying the Gal4 driver (*mef2*-Gal4 or *btl*-Gal4 crossed to *y w* flies) were used as controls in RNAi experiments. For rescue experiments, the fosmid clone FlyFos066598 (*Ejsmont et al., 2009*) containing the *Drosophila pseudoobscura Mmp1* homologue *GA18484* was integrated into the attP-9A/VK13 (76A2) landing site using PhiC31 integrase (*Bischof et al., 2007*).

## Genetic labeling of tracheal cell clones

For multicolor labeling of tracheal cells *btl*-Gal4 flies were crossed to *hsFlp::Pest;; UAS-FRT>STOP>FRT-myr::smGFP-HA_V5_FLAG* (*Nern et al., 2015*). L3 larvae were heat-shocked for 20 min at 37°C. The MARCM system (*Lee and Luo, 1999*) was used for clonal labeling of tracheal cells. For wild-type MARCM clones *y w hs-Flp122; FRT40A tub-GAL80; btl-GAL4 UAS-GFP* females were crossed to *y w hs-Flp122; FRT40A FRTG13; btl-GAL4 UAS-GFP* males. For *Mmp1* mutant clones *y w hs-Flp122; FRTG13 tub-GAL80; btl-GAL4 UAS-GFP* females were crossed to males carrying $Mmp1^2$, $Mmp1^{Q112}$ or $Mmp1^{Q273}$ mutations recombined on FRTG13 chromosomes. L3 larvae were heat-shocked for 2 hr at 37°C. Pupae were staged by collecting white pre-pupae and dissected 75 hr after puparium formation (APF) at 27°C.

## Flight test

Adult flies (one to five days after eclosion) were kept for four days at 30°C before testing for flight ability in a Plexiglas cylinder as described in *Weitkunat and Schnorrer (2014)*. We determined the percentage of flies that landed at the bottom immediately after throwing them into the Plexiglas cylinder.

## Immunostainings

For immunostainings of developing IFMs, white pre-pupae were staged at 27°C and dissected at the desired time APF. Pupae up to 60 hr APF were dissected according to the protocol for developing IFMs (*Weitkunat and Schnorrer, 2014*). Older pupae and adults (1–6 days after eclosion) were dissected according to the protocol for adult IFMs (*Weitkunat and Schnorrer, 2014*).

Primary antibodies were chicken anti-GFP (1:500; Abcam ab13970), mouse anti-DSRF (1:100; *Samakovlis et al., 1996*), chicken anti-HA (1:200; Abcam ab9111), anti-FLAG(M2) (1:1000; Sigma F1804), mouse anti-Dlg1 (1:200; DSHB 4F3), guinea-pig anti-LaminDm0 (1:1000; gift from Georg Krohne, University of Würzburg), rabbit anti-Laminin (1:1500; *Schneider et al., 2006*), mouse anti-MMP1 (mix of mouse monoclonal antibodies 3B8D12, 3A6B4, 5H7B11, 1:10/1:100/1:100; DSHB; *Page-McCaw et al., 2003*), rabbit anti-Perlecan (1:2000; *Schneider et al., 2006*), anti-ATP5A (1:100; Abcam ab14748). Goat secondary antibodies were conjugated with Alexa Fluor 488 or Dylight 488 (1:500; Life Technologies), Alexa Fluor 555, Alexa Fluor 568 (1:500; Life Technologies), Dylight 550 (1:500; Thermo Fisher) or Alexa Fluor 647 (1:500; Life Technologies). Tracheae in adult IFMs were visualized by their autofluorescence upon UV (405 nm) excitation. Phalloidin-TRITC (1:1000; Sigma) was used to stain F-actin and anti-HRP-Alexa Fluor 647 (1:100; Dianova) to label neurons.

## Transmission electron microscopy

Pupal and adult IFM samples were fixed at room temperature (RT) in 4% paraformaldehyde and 0.5% glutaraldehyde in 0.1 M phosphate buffer (PB) for at least 2 hr and were then transferred to 4° C overnight. On the next day, samples were fixed with 2% $OsO_4$ in 0.1 M PB for 1 hr on ice in the dark. Next, samples were washed five times using $ddH_2O$ and stained *en bloc* with 2% uranyl acetate (UA) for 30 min at RT in the dark. After five washes in $ddH_2O$, samples were dehydrated in an ethanol series (50%, 70%, 80%, 90%, 95%), each for 3 min on ice and twice in 100% ethanol for 5 min at RT. Following dehydration, samples were incubated twice in pure propylene oxide (PO) for 15 min and then transferred to an epon-PO mixture (1:1) to allow resin penetration over night. After removal of PO by slow evaporation over 24 hr, samples were embedded in freshly prepared epon (polymerization at 60°C for 24 hr). 90 nm sections were prepared on a Leica UC7 microtome and stained with 2% UA for 30 min and 0.4% lead citrate for 3 min to enhance contrast. Images were acquired with a Zeiss EM900 (80 kV) using a side-mounted camera (Osis).

## Light microscopy and image analysis

Stained specimens were imaged on a Zeiss LSM880 Airyscan confocal microscope or a Zeiss LSM710 confocal microscope. Z-projections were generated with Imaris (v9, Bitplane) using the '3D view' or with Fiji (GPL v2; *Schindelin et al., 2012*). Where indicated, images were acquired with the Airyscan detector and subjected to Airyscan processing.

## Live imaging of tracheal invasion into pupal IFMs

Live imaging of IFM tracheation was performed on a Leica SP8 confocal microscope using a 40x/1.3 NA oil immersion objective, resonant scanning mode and Hybrid Detectors. Pupae expressing *btl*-Gal4-driven palmitoylated mKate2 (palm-mKate2; *Caviglia et al., 2016*) to label tracheal cells and Myofilin-GFP (fTRG501; *Sarov et al., 2016*) to label muscles were prepared for live imaging 48 hr APF after staging at 27°C. The pupal case around the head and thorax was removed using forceps. Pupae were fixed on a coverslip with heptane glue and covered with a gas-permeable membrane (bioFOLIE 25, In Vitro System and Services, Göttingen, Germany) using a spacer of 0.5 mm. The dorsal-most DLMs were imaged from a dorsal view. DVMs were imaged from a lateral view. Time-lapse movies were recorded with z-stacks (100 µm thickness, 0.35 µm step size) acquired every 10 min over 14 hr.

## Analysis of tracheal invasion speed

To quantify the progress of IFM tracheation over time movies were processed with Fiji. A binary movie sequence of tracheal invasion was generated by first creating a maximum intensity projection, followed by histogram normalization (1% saturation) and auto thresholding (Huang filter). In a ROI of approximately 1680 $µm^2$ close to the medioscutal air sacs, changes in tracheal area fraction starting from 52 hr APF were assessed over time with the Time Series Analyzer Plugin.

## Quantification of tracheal density in IFMs

Confocal sections of tracheae in IFMs were taken below the muscle surface. Stacks of 9 µm thickness (step size 0.3 µm) were acquired along the selected myotube in different individuals. Tracheal branches were visualized in a myotube volume of approximately $3 \times 10^4$ $µm^3$. After average intensity

projection, background subtraction (sliding paraboloid, radius 15), median filtering (radius 2), histogram normalization (1% saturation) and manual thresholding were performed in Fiji to generate binary images of tracheal branches. These binary images were used to determine the fraction of myotube area occupied by tracheal branches.

### Quantification of tracheal branch points per myotube volume

To determine the number of tracheal branch points in a myotube volume of approximately $3 \times 10^4$ $\mu m^3$ the same image raw data as described above were used. 3D binary images of tracheal branches were generated in Fiji by background subtraction (sliding paraboloid, radius 15), median filtering (radius 2), histogram normalization (1% saturation), Gaussian blur (sigma 0.2) and manual thresholding. The 3D binary images were subjected to the Skeletonize 3D plugin of Fiji to count the number of branches and branch points.

### Analysis of tracheal branching angles

The angles of branches extending from tracheal cell bodies on the IFM surface were determined by superimposing a circle with defined radius over a tracheal branch tree such that the circle covers the branch tree and the stalk. The intersections of tracheal branches with the circumference of the circle were recorded and plotted.

### Morphometry of tracheal cells

Z-stacks of single MARCM-labeled tracheal cells were acquired with a step size of 0.5 μm. Brightness correction along z was applied during image acquisition to visualize finest branches deep inside the myotube. 3D binary images of tracheal branches were generated in Fiji by background subtraction (sliding paraboloid, radius 15), median filtering (radius 2) and manual thresholding, and were segmented in Imaris. The Surface tool of Imaris was used to determine the cellular volume. The Filament tracer tool was used to segment branches and to extract the sum of branch length, total number of branch points and terminal points, branch straightness, branch orientation angle and branching angle. Automated filament tracing was adjusted manually for correctness.

### Analysis of mitochondrial morphology

Airyscan images of IFM mitochondria were acquired with a step size of 0.3 μm. Mitochondria were segmented using the Surface tool of Imaris with 'split touching objects' enabled. Analysis of segmented mitochondria was performed using the Vantage tool of Imaris.

### Statistics and reproducibility

For phenotypic analyses, sample size (n) was not predetermined using statistical methods, but was assessed by taking into account the variability of a given phenotype, determined by the standard deviation. Experiments were considered independent if the specimens analyzed were derived from different parental crosses. During experiments investigators were not blinded to allocation. For statistical analysis, the Kolmogorov-Smirnov test was applied, which does not require assumptions on the type of data distribution.

### Quantification of mitochondria and tracheae in TEM sections

For each developmental time point, at least three to five TEM cross-sections were analyzed for the area of mitochondria and tracheal lumen, respectively, by manually tracing the outlines of mitochondria and of tracheal lumen in ImageJ. The mean areas of mitochondria and of tracheal lumen, respectively, are shown in *Figure 2—figure supplement 1I*.

## Acknowledgements

We thank Anna Körte for help with the RNAi screen, Aynur Kaya-Çopur for introducing us to flight muscle dissection and for providing samples for electron microscopy analysis, and Manuel Hollmann and Mylène Lancino for comments on the manuscript. We thank Markus Affolter, Stefan Baumgartner, Andrea Page-McCaw and Mirka Uhlirova for providing fly stocks and reagents. We are grateful to Christian Lehner for discussions and support at the Institute of Molecular Life Sciences (University

of Zurich) during the first phase of this project. We are grateful to Christian Klämbt for discussions and for providing access to facilities at the Institute of Neuro- and Behavioral Biology (University of Münster).

## Additional information

### Funding

| Funder | Grant reference number | Author |
|---|---|---|
| Boehringer Ingelheim Fonds | Fellowship | Julia Sauerwald |
| European Molecular Biology Organization | Young Investigator Programme | Frank Schnorrer |
| H2020 European Research Council | 310939 | Frank Schnorrer |
| Centre National de la Recherche Scientifique | | Frank Schnorrer |
| Excellence Initiative Aix-Marseille University AMIDEX | ANR-11-IDEX-0001-02 | Frank Schnorrer |
| LabEX-INFORM | ANR-11-LABX-0054 | Frank Schnorrer |
| Fondation Bettencourt Schueller | | Frank Schnorrer |
| Deutsche Forschungsgemeinschaft | CRC 1348 | Stefan Luschnig |
| Deutsche Forschungsgemeinschaft | CRC 1009 | Stefan Luschnig |
| Schweizerischer Nationalfonds zur Förderung der Wissenschaftlichen Forschung | SNF 31003A_141093_1 | Julia Sauerwald Till Matzat Stefan Luschnig |
| Deutsche Forschungsgemeinschaft | Cluster of Excellence | Julia Sauerwald Wilko Backer Till Matzat Stefan Luschnig |

The funders had no role in study design, data collection and interpretation, or the decision to submit the work for publication.

### Author contributions

Julia Sauerwald, Data curation, Formal analysis, Validation, Investigation, Visualization, Methodology, Writing—original draft, Writing—review and editing; Wilko Backer, Validation, Investigation, Visualization, Methodology; Till Matzat, Data curation, Validation, Investigation, Visualization, Methodology; Frank Schnorrer, Conceptualization, Resources, Supervision, Investigation, Methodology, Writing—review and editing; Stefan Luschnig, Conceptualization, Resources, Supervision, Funding acquisition, Validation, Investigation, Writing—original draft, Project administration, Writing—review and editing

### Author ORCIDs

Frank Schnorrer  https://orcid.org/0000-0002-9518-7263
Stefan Luschnig  https://orcid.org/0000-0002-0634-3368

### Decision letter and Author response

Decision letter https://doi.org/10.7554/eLife.48857.028
Author response https://doi.org/10.7554/eLife.48857.029

# Additional files

## Supplementary files

• Supplementary file 1. Quantitative analysis of IFM tracheal terminal cell morphology. Volume, sum of branch length, and total number of branch points and terminal points were extracted from 31 individually marked (MARCM clones) segmented terminal tracheal cells in wild-type pupae (75 hr APF).
DOI: https://doi.org/10.7554/eLife.48857.022

• Supplementary file 2. RNAi screen to identify genes required for IFM tracheal invasion. Columns include gene identifiers and gene names of selected candidate genes, details about the RNAi lines used to knock down candidate genes, and the results of the screen for genes required for flight ability and tracheal invasion into myotubes.
DOI: https://doi.org/10.7554/eLife.48857.023

• Supplementary file 3. Statistics Reporting Table.
DOI: https://doi.org/10.7554/eLife.48857.024

• Transparent reporting form
DOI: https://doi.org/10.7554/eLife.48857.025

## Data availability

All data generated or analysed during this study are included in the manuscript and supporting files.

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
