## [Decision Letter]

Thank you for submitting your article "Mmp1 modulates invasive behavior of tracheal branches during ingression into *Drosophila* flight muscles" for consideration by *eLife*. Your article has been reviewed by two peer reviewers, and the evaluation has been overseen by K VijayRaghavan as the Senior and Reviewing Editor. The following individual involved in review of your submission has agreed to reveal their identity: Christos Samkovlis (Reviewer #2).

The reviewers have discussed the reviews with one another and the Reviewing Editor has drafted this decision to help you prepare a revised submission.

Summary:

The study by Sauerwald and colleagues describes the mechanism of how tracheae invade and colonize *Drosophila* Salm-specified, fibrillar flight muscles during pupal development. Combining genetics with high-end imaging of live and fixed samples, and rigorous image analyses the authors provide a detailed description of the invasion process which requires directional migration of trachea inside the myotubes and spreading of the tracheal branches until they fill the myotubes. The authors characterize the dynamic behavior of the growth-cone-like structures at branch tips and the differential composition of the basal membrane along the tracheal branches. They show that Laminin is deposited along the entire tracheal branch while Perlecan and Collagen IV are restricted to the tracheal stalks. They provide causal genetic evidence for the role of MMP1 in the invasion process and implicate MMP1 in the dynamics of growth-cone-like branch tips through the remodeling of the ECM component Collagen IV. Based on the cell culture experiments, the authors propose an additional role for trachea-derived MMP1 in generating a chemoattractant gradient of FGF/Bnl around the migrating tracheal branches. The study provides novel insights into the mechanisms of muscle tracheation, which involves the paradigm of controlled cell invasion and intercellular crosstalk between different tracheal cells and the invaded myotube. The manuscript is well written and understandable, authors present a set of well-conducted, suited, high-quality experiments followed by rigorous data analyses and visualization generating a solid foundation to support the major conclusions of the study.

Essential revisions:

1) Although they detected the proximity of the trachea to the muscle mitochondria they could not detect the spectacular wrapping of tracheal projections around these organelles. In this context, it would be interesting for the reader to provide a quantification of the EM analysis.

2) To investigate the genetic determinants of the muscle-tracheal interactions, the authors performed small scale screen of preselected RNAi strains inactivating genes either in flight muscles or in the trachea. They show that the transcription factor Salm is required in the flight muscle for the attraction and penetration of tracheal cells. Conversely, Bnl signalling and Mmp1 are required in the tracheal cells for proper muscle invasion. They focused on the genetic analysis of Mmp1 in the process and they convincingly showed that Mmp activity is required for ECM remodelling. Interestingly collagen was the major target of Mmp1, compared to other ECM components like perlecan or laminin. The genetic analysis of Mmp1 inactivation suggested that additional roles for Mmp1 apart from the degradation of collagen. The authors tried to investigate the possibility that Bnl could be a proteolytic substrate of Mmp1. They overexpressed combinations of Bnl, Btl, and Mmp1 in S2 cells and detected Bnl and Mmp1 in the cellular extracts (or in culture medium?). They show a Western blot where they detect reduced Bnl when Mmp1 is coexpressed and conclude that Mmp1 cleaves and degrades Bnl. This experiment is poorly controlled and not necessary in the context of this work. If the authors want to argue that for specific cleavage of Bnl by Mmp1 they should also show in parallel the levels of Btl and another secreted protein like Delta or Notch in the same experiment.

3) The study refers too many times to "data not shown". The data that are relevant should be presented and "data not shown" be avoided"

4) Did authors try to assess the MMP1 levels along the branches? Could the differential distribution of MMP1 support their proposed role for MMP1 in generating the Bnl gradient?

5) Is MMP1-induced Bnl degradation in the cell culture system associated with reduced downstream signaling?

6) The authors might comment on the frequency of mmp1LOF clones in time points earlier than 75h APF, otherwise, the last paragraph of the subsection “MMP1 is required in tracheal cells for invasion into myotubes”,, remains very speculative.

7) Concerning the Bnl/Btl signaling, they could look at ERK activation as potential evidence for target perturbation.

---

## [Author Response]

Essential revisions:1) Although they detected the proximity of the trachea to the muscle mitochondria they could not detect the spectacular wrapping of tracheal projections around these organelles. In this context, it would be interesting for the reader to provide a quantification of the EM analysis.

We appreciate the reviewer´s comment. We carefully revisited our set of TEM data. In no case did we observe wrapping of tracheoles around mitochondria. To our knowledge, such wrapping has previously been reported only in studies based on dye infiltration experiments, which are prone to artifacts. However, in adult IFMs we found many cases of tracheoles that appeared to be partially enwrapped by large sleeve-like mitochondria. This is likely to be explained by the dramatic increase in mitochondrial size during late-pupal IFM development, which is accompanied by extensive fusion of mitochondria, giving rise to giant sleeve-like mitochondria around tracheal tubes. To support this point, we revisited our TEM images of IFM cross-sections at 32h h APF, 48 h APF and in adults, and analyzed the size of mitochondria and tracheal lumina. The cross-sectional area of mitochondria increased with developmental time, most dramatically between 48 h APF and adulthood, lending support to our hypothesis that growth and/or fusion of mitochondria results in enwrapping of tracheoles. We show the quantification of the EM data in a new panel (I) added to the figure supplement to Figure 2 (Figure 2—figure supplement 1I).

2) To investigate the genetic determinants of the muscle-tracheal interactions, the authors performed small scale screen of preselected RNAi strains inactivating genes either in flight muscles or in the trachea. They show that the transcription factor Salm is required in the flight muscle for the attraction and penetration of tracheal cells. Conversely, Bnl signalling and mmp1 are required in the tracheal cells for proper muscle invasion. They focused on the genetic analysis of Mmp1 in the process and they convincingly showed that Mmp activity is required for ECM remodelling. Interestingly collagen was the major target of Mmp1, compared to other ECM components like perlecan or laminin. The genetic analysis of mmp1 inactivation suggested that additional roles for mmp1 apart from the degradation of collagen. The authors tried to investigate the possibility that Bnl could be a proteolytic substrate of Mmp1. They overexpressed combinations of Bnl, Btl, and Mmp1 in S2 cells and detected Bnl and Mmp1 in the cellular extracts (or in culture medium?). They show a Western blot where they detect reduced Bnl when mmp1 is coexpressed and conclude that Mmp1 cleaves and degrades Bnl. This experiment is poorly controlled and not necessary in the context of this work. If the authors want to argue that for specific cleavage of Bnl by Mmp1 they should also show in parallel the levels of Btl and another secreted protein like Delta or Notch in the same experiment.

We appreciate the reviewer´s comment regarding the cell culture experiments. We detected Bnl and Mmp1 proteins from cellular extracts as well as from the culture medium, indicating that both proteins were secreted by the cells. However, the experiment in Figure 6 shows an immunoblot of cellular extracts only. To address the specificity of Mmp1 activity towards Bnl as a possible proteolytic substrate, we tried to detect as an additional potential substrate the FGF receptor Breathless (Btl), but we were unable to obtain specific signals using an available rabbit anti-Btl antiserum.

To validate the results from the tissue culture experiments in vivo, we made extensive efforts to visualize the endogenous Bnl protein in IFMs. However, using the available reagents (anti-Bnl antiserum, flies expressing GFP-tagged Bnl from the endogenous *bnl* locus), we were unable to detect the Bnl protein in whole-mount preparations of pupal or adult flies. Given these results, we agree with the reviewer´s statement that the cell culture experiments do not contribute to the main conclusions of the paper. We therefore decided to remove the section describing these results, along with Figure 6, from the manuscript.

3) The study refers too many times to "data not shown". The data that are relevant should be presented and "data not shown" be avoided".

In the revised manuscript we present relevant data that was previously referred to as “data not shown”:

Subsection “IFM mitochondria enwrap tracheal cells, but not vice versa”: We added two panels to Figure 2—figure supplement 1L, M to show the quantitative analysis of mitochondria previously referred to as “data not shown”.

Subsection “salm-dependent flight muscle fate is required for tracheal invasion”: We now show the innervation of trachealess muscles of flies with muscle-specific knockdown of *bnl* previously referred to as “data not shown” (new panels E and F in Figure 3—figure supplement 1).

Discussion, fifth paragraph: The distribution of tracheoles in IFMs of *amph* mutants is shown in Figure 5—figure supplement 1G, H. We removed the reference to the number of tracheoles in *amph* mutants and the statement “data not shown”.

4) Did authors try to assess the MMP1 levels along the branches? Could the differential distribution of MMP1 support their proposed role for MMP1 in generating the Bnl gradient?

We tried hard to assess the distribution of Mmp1 protein using three different anti-Mmp1 monoclonal antibodies (3B8, 3A6, 5H7; Page-McCaw et al., 2003). However, although we tested various different fixation protocols (paraformaldehyde, glutaraldehyde, Bouin´s fixative), we were not able to obtain specific signals with these antibodies in immunostainings of whole-mount preparations of pupal or adult IFMs.

We also made considerable efforts to generate an endogenously tagged version of Mmp1. We inserted a V5 epitope tag into the MMP1 hinge region by targeting the endogenous *mmp1* locus using CRISPR-mediated genome editing. However, while V5-tagged Mmp1 protein was detectable in protein extracts by immunoblot, no specific signal was visible in anti-V5 immunostainings of whole-mount preparations of pupal or adult IFMs. We therefore did not include these results in the manuscript.

Moreover, we also analyzed flies carrying a genomic fosmid clone in which MMP1 has been fused at its C-terminus to superfolder GFP and FLAG, TY1, and V5 epitope tags (fTRG145; Sarov et al., 2016; VDRC). Although we were unable to detect MMP1-GFP signals in living specimens, we detected specific signals in whole-mount preparations stained with an anti-GFP antibody. Discrete MMP1-GFP staining was visible on the IFM-associated tracheal air sac epithelium and along tracheal branches inside IFM myotubes, consistent with our finding that MMP1 is required for normal invasive behavior of tracheal branches inside IFM myotubes. We describe these findings in the text (subsection “MMP1 is required in tracheal cells for invasion into myotubes”) and show the data in a new supplementary figure (Figure 4—figure supplement 1).

Addressing the reviewer’s question regarding the role of MMP1 in generating a Bnl gradient in IFMs, we tried extensively to detect Bnl FGF in situ, either in fixed specimens using anti-Bnl antibodies or in living pupal or adult specimens expressing a GFP-tagged knock-in allele of *bnl* (Du L, Sohr A, Yan G and Roy S (2018) Feedback regulation of cytoneme-mediated transport shapes a tissue-specific FGF morphogen gradient. *eLife* 7). However, neither of these approaches revealed specific signals in pupal or adult IFMs. Thus, based on these results we cannot provide more direct evidence for a role of MMP1 in modifying the distribution of Bnl FGF in developing IFMs.

5) Is MMP1-induced Bnl degradation in the cell culture system associated with reduced downstream signaling?

To test whether signaling downstream of Bnl is reduced upon MMP1-induced degradation of Bnl we analyzed MAPK activation by detecting dpERK on immunoblots (Author response image 1). This did not reveal clear changes in dpERK levels between cells transfected with MMP1 and control cells (Author response image 1; compare lanes 3 and 4). Given these results, we agree with the reviewer´s statement that the cell culture experiments do not contribute to the main conclusions of the paper. We therefore decided to remove the section describing these results, along with Figure 6, from the manuscript.

**Author response image 1. respfig1:** Immunoblot of cellular protein extracts of S2 cells transfected as indicated on top.

6) The authors might comment on the frequency of mmp1LOF clones in time points earlier than 75h APF, otherwise, the last paragraph of the subsection “MMP1 is required in tracheal cells for invasion into myotubes”, remains very speculative.

We have no data on the occurrence of *mmp1*^LOF^ mutant clones at stages earlier than 75h APF. We are aware that the conclusion is based on negative evidence and hence has to remain speculative. To take this into account we changed “*suggests*” to “*is consistent with*” in the conclusion of the paragraph, which now reads:

“Although we cannot exclude the presence of additional cell-lethal mutations on the Mmp1 mutant chromosomes, the absence of homozygous Mmp1 mutant clones from IFMs is consistent with an essential cell-autonomous requirement of MMP1 in IFM tracheation.”.

7) Concerning the Bnl/Btl signaling, they could look at ERK activation as potential evidence for target perturbation.

Please see above under 5.